# Evaluation of the Likelihood of Establishing False Codling Moth (*Thaumatotibia leucotreta*) in Australia via the International Cut Flower Market

**DOI:** 10.3390/insects13100883

**Published:** 2022-09-28

**Authors:** Xingyu Li, Robert N. Emery, Grey T. Coupland, Yonglin Ren, Simon J. McKirdy

**Affiliations:** Harry Butler Institute, Murdoch University, Perth, WA 6150, Australia

**Keywords:** biological invasion, *Thaumatotibia leucotreta*, species distribution model, Maxent, climate suitability, habitat suitability

## Abstract

**Simple Summary:**

False codling moth is a polyphagous pest that could be introduced and establish in Australia. This moth may threaten many commercial horticultural and agricultural production crops in Australia by damaging a wide range of agricultural crops including avocados, guavas, peaches, citrus, grapes, cotton, roses and some solanaceous crops. Since false codling moth has caused considerable financial loss to rose growers through damage to agricultural crops and horticultural products as well as the costs associated with its control and eradication in some African countries, it is prudent to predict likely establishment regions in Australia. The predictions were generated using a Species Distribution Model named Maxent. Results not only considered the climatic suitability but also overlaid the prediction maps with presence of susceptible hosts. The predictions indicated a range of near-ocean regions across Australia that were potentially suitable habitats. These results provided preliminary insight into the potential for false coding moth to establish in Australia. This research is valuable as the economic impact of this pest could be serious, with prediction and early detection key to preventing the establishment and spread of false codling moth across Australia.

**Abstract:**

Kenya and some other African countries are threatened by a serious pest *Thaumatotibia leucotreta* (Meyrick) (Lepidoptera: Tortricidae), the false codling moth. The detection of *T. leucotreta* is quite difficult due to the cryptic nature of the larvae during transportation and is therefore a concern for Australia. This insect is a known pest of agriculturally important crops. Here, Maxent was used to assess the biosecurity threat of *T. leucotreta* to Australia. Habitat suitability and risk assessment of *T. leucotreta* in Australia were identified based on threatened areas under suitable climatic conditions and the presence of hosts in a given habitat. Modeling indicated that Australia is vulnerable to invasion and establishment by *T. leucotreta* in some states and territories, particularly areas of western and southern Australia. Within these locations, the risk is associated with specific cropping areas. As such, invasion and establishment by *T. leucotreta* may have serious implications for Australia’s agricultural and horticultural industries e.g., the fruit and vegetable industries. This study will be used to inform the government and industry of the threat posed by *T. leucotreta* imported via the cut flower industry. Targeted preventative measures and trade policy could be introduced to protect Australia from invasion by this pest.

## 1. Introduction

Australia is a net agricultural exporter, with the gross value of agricultural production exceeding $66 billion in 2020–2021 [1]. Agricultural exports play an important part in the Australian economy, with 70% of Australian agricultural production exported annually due to the relatively low Australian population and high production [2]. As such, Australia’s agriculture industry must be protected from invasive pests that might threaten agricultural exports.

At the same time, Australia is an importer of horticultural products. From 2019–2020, the Australian export of fresh cut-flower produce was valued at AUD8.4 million while import produce was valued at AUD74.2 million [3]. Kenya is one of the largest exporters of cut flowers to Australia, supplying Australia with 20% of their imported cut flowers: 80,557,649 units valued at USD12.7 million in 2019 [4]. Rose (Rosales: Rosacea) imports comprise the largest volume of consignments (approximately 34%) of the total volume of fresh cut-flowers imported during the last decade [5].

*Thaumatotibia leucotreta* (Meyrick) (Lepidoptera: Noctuidae), also known as false codling moth, is a European Union quarantine pest. *T. leucotreta* is a known pest of agriculturally important crops, including avocados, guavas, peaches, citrus, grapes, cotton, some solanaceous crops and roses [6,7,8]. In recent years, *T. leucotreta* has been deemed a priority pest by many countries, with pathways of human-assisted spread, such as passenger luggage and imported fresh products [9]. *T. leucotreta* is native to sub-Saharan Africa [7] and has been established in two non-indigenous regions, Western Cape of South Africa [10] and Israel [11]. Currently, this species is a major pest of economic concern in Africa, and endemically established across the African land mass, including Angola, Ghana, Ethiopia, Kenya, South Africa and some island countries, including Madagascar and Côte d’Ivoire [7]. Additionally, *T. leucotreta* is present but restricted in Israel [6,7] and was temporarily detected but successfully eradicated from glasshouses in Netherlands (2014) [12]. *Thaumatotibia leucotreta* is considered a significant economic pest in Kenya and other African counties as its impact on agricultural areas threatens local and export crop production including vegetables, fruits and flowers, as well as reducing the export of capsicums, roses and avocados to the EU and around the world. The pest also has a negative impact on Kenya citrus production [13]. The economic impact involves yield losses of crops, cost of control and management of the pests and negative environmental and health impacts associated with the use of insecticides. More recently, *T. leucotreta* is now classified as not only a production pest, but also regulated as a phytosanitary pest with declared ‘zero tolerance’ [14]. This could involve ongoing economic losses, impact on reputation, customer loyalty and even import licensing [15]. *Thaumatotibia leucotreta* is typically polyphagous and can cause feeding damage to over 107 host species and subspecies, both cultivated and wild [6,7,9,16,17,18]. Although *T. leucotreta* has low natural dispersal ability across the landscape, it is able to move long distances via human-mediated pathways [19]. Many hosts can be classified as “green bridges”, allowing pests and diseases access to span cropping seasons [20,21]. This facilitated mobility and host availability increases the challenge of control as the pest moves between crops grown in the same field and between fields [21,22].

Damage caused by *T. leucotreta* can include a range of symptoms, such as chewing, exit holes and frass eventually resulting in yield reduction [23], and infested plants are more susceptible to infection by fungi and bacteria [18]. A further challenge associated with managing *T. leucotreta* is that infestation symptoms are often difficult to detect. In the case of roses, the flower buds can appear to be undamaged during early infestation [9].

In 2004, the European Commission declared *T. leucotreta* as a Union quarantine pest listed in Part A of Annex II of Commission Implementing Regulation ((EU)2019/2072) and listed as priority pests in Commission Delegated Regulation ((EU)2019/1702) [6] and the pest has caused a considerable financial loss to Kenya’s rose growers through shipment rejection [8]. Kenya remains a major source of *T. leucotreta* in the export of cut flowers [24]. There has been an increasing number of *T. leucotreta* interceptions in the EU over recent years with the most noticeable interception increases reported in roses [7]. The interception statistics for the EU showed no detections of *T. leucotreta* in cut flowers from Kenya between 2005 and 2017, however, there were 37 detections in 2018, 39 in 2019, 51 in 2020, 50 in 2021 and 9 until June of the current year, 2022 [25].

There is no evidence that *T leucotreta* can enter diapause [16,26,27]. Boardman et al. [28], however, reported that larvae of *T. leucotreta* may pass into a “chill coma” at temperatures of 3 °C–7 °C, becoming dormant, and ceasing feeding and other activity. However, Moore et al. [29] obtained 100% mortality of *T. leucotreta* at temperatures 1, 2 and 3 °C for 19, 20 and 24 days. Huisamen et al. [30] reported that 4 °C was not likely to affect the *T. leucotreta* fitness (spontaneous behavior and flight ability) based on simulating lower-temperature transportation. The shipping temperatures for cut flowers sent to Australia is 4 °C [31]. To ensure the maximum vase life of the fresh cut-flowers, those imported products are often transported via air freight at temperatures between 1 °C and 4 °C [31]. As such, the pest is unlikely to die, instead it will limit its activity during export.

While *T. leucotreta* is a tropical pest, it is known to be active during winter nights in South Africa, indicating that the pest has a greater tolerance to low temperatures compared with the codling moth, *Cydia pomonella* [32]. Moreover, European and Mediterranean Plant Protection Organization (EPPO) [6] identified that *T. leucotreta* is mostly considered to be particularly vulnerable under continuing cold conditions, however, regions with a greater diurnal fluctuation range tends to favor the pest overwintering even under cold conditions. As the pest may be able to complete more than one generation during the summer period, this may allow transient populations to survive [6]. This suggests that *T. leucotreta* may also be a threat to more temperate countries. Venette [17] forecasted that around 20% of continental USA is a potentially suitable habitat for *T. leucotreta*, particularly in the southern and south-western United States [33]. Given its frequent interception in Europe [25] and United States [10,33], Australia has identified *T. leucotreta* as a high risk as it is assumed to be capable of establishing itself in Australia and is considered a serious biosecurity threat, particularly to a wide range of crops across tropical and sub-tropical areas [23].

Given the threat posed by *T. leucotreta*, determining the accurate geographic area that is most likely threatened by potential establishment is essential for effective monitoring and management. Species Distribution Models (SDMs) are a useful tool for predicting the potential distribution of species [34] based on an estimation of the relationship between known species distribution data (occurrence records) and environmental data (environmental variables/predictors) of the area under consideration [35]. In this study, Maxent, based on the principle of maximum entropy [36], was selected to predict the potential distribution of the species. Maxent is commonly used to predict the areas susceptible to pest species using presence-only data [37]. Compared to other techniques, Maxent shows higher robustness [38] as well as working well on rare species with a small sample size [39].

Climate change is another concern with SDMs prediction in the future. Climatic changes cause the change in greenhouse gases which may affect the distribution of the pest [40]. Levi-Zada et al. [41] mentioned climate change as the possible reason for recent outbreaks of *T. leucotreta* in pomegranate plantations in Israel. The global mean surface temperature is becoming warmer along with the global long-term warming trend identified by NASA’s Goddard Institute for Space Studies (GISS) [42]. The increasing global temperature has the potential to alter the suitable area for *T. leucotreta.*

In light of the biosecurity concern of an incursion by *T. leucotreta* into Australia and the potential for its establishment, *T. leucotreta* represents a clear and present threat to Australia’s agricultural and horticultural sectors. As post-border detections of *T. leucotreta* inevitably increase, this research seeks to provide information to biosecurity managers, industry and government departments on the potential regions that are likely to support the infestation and establishment of this pest with the specific aims: (i) to estimate the potential geographic distribution of *T. leucotreta* in Australia, along with the strength of habitat suitability and (ii) to identify the most important environmental predictors that drive the establishment range of *T. leucotreta*.

This research aims to guide development of early detection methods, monitoring systems and other phytosanitary interventions that will assist with the biosecurity management of *T. leucotreta* (False codling moth).

## 2. Materials and Methods

### 2.1. Thaumatotibia leucotreta Occurrence Data

The occurrence data for *T. leucotreta* were collected from currently available distribution records obtained from a range of sources: Global Biodiversity Information Facility (GBIF; https://www.gbif.org/ (accessed on 16 October 2020)); Centre for Agriculture and Bioscience International (CABI; https://www.cabi.org/cpc (accessed on 17 October 2020)); and European and Mediterranean Plant Protection Organization (EPPO; https://www.eppo.int/ (accessed on 17 October 2020)).

The total 89 occurrence points used for the modeling of *T. leucotreta* in this study are shown in Figure 1. These data, based on current distribution utilized for *T. leucotreta*, were managed using a bias grid file created by ‘ENMeval’ package [43] in RStudio Version 1.4.1106 (PBC, MA, Boston), which was initially built for Maxent models [44,45]. This was to ensure the background data points had the same sampling bias with presence locations. This grid file was used in order to reduce the potential spatial autocorrelation. Spatial autocorrelation is a problem caused by spatial sampling bias, and generally occurs in ecological research data because of a lack of independence between pairs of observations within a specific geographical space [46]. This problem may reduce model performance [47] and have an adverse impact on model quality [39,48].

### 2.2. Environmental Data

Climate is one of the most important factors driving the distribution of phytophagous insects [49]. In most SDM studies, the environmental variables are also known as predictors, covariates or inputs in statistical terms [37]. Nineteen bioclimatic variables including temperature and precipitation are commonly employed for modeling SDMs. For this study, 19 bioclimatic variables of WorldClim Version 2 were downloaded from the WorldClim dataset at 10 arc-minutes spatial resolution (Table 1), with a temporal boundary between 1970 and 2000 [50]. This study also assumed ‘degree days above 10 °C’ as a threshold for *T. leucotreta* population development as eggs of *T. leucotreta* have a higher mortality rate at temperatures below 13 °C [51] and cessation of hatching was noted at 10.6° C [16]. This refers to Kumar’s [52] research of using the variable ‘Degree days’ as a species-specific phenology variable that may improve the predictive power of correlative niche models [53,54]. As such, this variable was calculated by raster calculator through ArcToolbox of ArcGIS [55], according to Nugent’s [56] study. The 19 Bioclimatic variables as well as ‘Degree days above 10 °C’, were included based on the biological requirement of *T. leucotreta* for temperatures above 10 °C for egg laying and an optimal temperature for colonization of 25 °C [16,32,51].

Collinearity is a common problem when statistical models are used to estimate the relationship between one response variable and a set of predictor variables [57]. To reduce collinearity, this study used the Spearman rank-order to remove correlations in variables with higher pairwise correlation coefficients (|r| > 0.8) bolding (Appendix A
Table A1). Distilling predictors in advance by using ecologically relevant environmental variables is a preferred approach [34], i.e., variables with less biological relevance were excluded from high collinearity variables for this study. In this study, 13 variables were eliminated and seven variables (Table 1, highlighted with a tick) were used in modeling.

### 2.3. Maxent Modeling

Models were processed using the ‘Maxent’ platform (Version 3.4.1). The two parameters that have the greatest influence on model accuracy are ‘feature classes’ and ‘regularization multiplier’ (RM) [58]. Feature classes include Linear (L), Quadratic (Q), Product (P), Threshold (T) and Hinge (H) [41]. Regularization parameters work to smooth the model and prevent it from ‘overfitting’ [44,58]. As well as presence data (occurrence records), 10,000 background points were randomly selected as pseudo-absence data. This approach used the commonly adopted procedure ‘K-fold cross-validation’ (CV), which efficiently uses all data for validation and allows for uncertainty in predictions [59,60]. The results were generated with 10 replicates across the models. Logistical output was used for ease of interpretation and estimation of the probability of occurrence conditioned by environmental variables [58]. In addition, logistical output is robust to unknown prevalence [37,58].

### 2.4. Model Performance and Preferred Model Selection

This study considered both threshold-independent metrics and threshold-dependent metrics to evaluate and select the optimal model. Area under the curve (AUC) of the receiver operating characteristic (ROC) curve [61] is a threshold-independent metric that characterizes the performance of models in many applied SDM research [44,60]. The AUC value estimates whether the presence points (locations) have greater habitat suitability values (HSV) than a random selection of pseudo absences from the study area [62]. The value of AUC varies between 0 and 1, with the random prediction corresponding to the value of 0.5 [44]. Values under 0.5 indicate poor performance, i.e., worse than a random model, and as model performance increases to 1, approaching a perfect fit model [63]. Akaike’s information criterion (AIC) [64] is another threshold-independent metric used to evaluate the models generated by the ‘ENMeval’ package [43]. The model should be selected with the minimum AIC value at zero [65]. The threshold-dependent metric used in this study is ‘omission rate at minimum training presence threshold’ (OR), and ‘omission rate at 10% training presence threshold’ (OR10). The OR and OR10 values approximate to 0 and 1. 1, respectively, and indicate better performance of the model [47]. The selection of the optimal model followed a sequence of lowest AIC value at zero, lower OR and OR10 values, as well as higher AUC values.

The final optimal model chosen for this study included the threshold dependent metric ‘Delta AIC’ at 0 with a good AUC value (0.954 ± 0.020). The threshold independent metric of the optimal model shows a low omission rate of 0.03 as well as a low value of omission rate (10%) of 0.15. The optimal model used in this study included six climatic variables: bio4, bio6, bio7, bio13, bio17, bio19 and DDA10 (degree days > 10 °C). All feature classes: linear, quadratic, product, threshold and hinge (LQPTH) were included. The ‘regularization multiplier’ (RM) value was set at 3.

### 2.5. Prediction of Habitat Suitability and Identification of Key Variables

Based on the output of the most optimistic model, ArcMap 10.6 [55] was used to transform the logistic output into habitat suitability maps based on one of the best thresholds ‘sensitivity-specificity sum maximization’ [66]. Liu [67] also identified that using Maxent with ‘sensitivity-specificity sum maximization’ has higher performance for common species with presence-only data. The habitat suitability maps were classified into four layers using the Spatial Analyst SDMToolbox 2.0 [68] of ArcMap. Four classes of suitability were used based on Jenks Natural Breaks Classification [69]: Unsuitable (0–0.1), Low (0.1–0.2), Moderate (0.2–0.4) and High (0.4 to 0.8).

To identify the potential for *T. leucotreta* to establish in Australia, the areas of potential species invasion considered both climatic habitat suitability and host range [9,18], as well as the pest’s ability to disperse across susceptible hosts [9]. Given that *T. leucotreta* is a polyphagous pest, its host range could be much wider than currently known. This research incorporates the extent of possible hosts using the horticultural and agricultural land use data from ‘Australian land use and management classification’ [70]. According to the classification, the prediction considered normal, irrigated and intensive horticulture, as well as cropping areas with known susceptible, or potentially susceptible hosts. The overlay map was generated by ArcMap, with the climatically suitable habitat of *T. leucotreta* superimposed on host land use. In addition, the potential establishment area (hectares) of each state or territory was estimated using ArcMap and Excel.

The Maxent model’s internal Jackknife test and contribution percentage were both used to identify the relative importance of different environmental variables and the relative contribution that each variable made to the model [59,70]. Jackknife test could estimate the variance and bias of large populations via the regularized training gain [71]. In addition, response curves were used to analyze the predicted probability of the species presence response to environmental variables [59].

## 3. Results

### 3.1. Predicted Habitat Suitability of Areas for Potential Establishment by Thaumatotibia leucotreta

The optimal model predicted climatically suitable areas for *T. leucotreta* in Australia (Figure 2). The central area of Australia was determined as an unsuitable habitat, because it is mostly desert with virtually no agricultural areas. The most suitable areas for *T. leucotreta* were generally located in the coastal regions of each state and territory (Figure 2). The most highly suitable areas are clustered in the south-west of Western Australia (WA), including temperate agricultural areas (Shire of Augusta-Margaret River, Shire of Albany and catchment shires or cities in southern parts of the Wheatbelt). The south-east part of South Australia (SA), including agriculture areas in temperate southern Australia (Fleurieu Peninsula and Kangaroo Island), is also highly suitable. Additionally, there were a few scattered areas of high suitability in the tropical northern part of the Northern Territory (NT), northern and eastern part of Queensland (Qld), the temperate eastern part of New South Wales (NSW), as well as the northern part of Tasmania (Tas.). The majority of moderate and low suitability areas were found in QLD and NSW along the coast. Moderate and low suitability areas also covered southern Victoria (Vic.), the eastern coastal area of SA, the northern part of NT and some northern regions of Tasmania.

The overlay map (Figure 3) combines areas of climatic suitability with host crop production areas and highlights the high suitability areas in WA and SA (Figure 4). In WA, the total measurement of highly suitable areas is around 89,670 hectares (ha) (Table 2). The potential invasion and establishment extent includes the agricultural areas in parts of temperate south-western Australia (Augusta, Albany and Manjimup) (Figure 3, Table 2), with the possible cultivated hosts embracing perennial, seasonal irrigated and intensive cropping and horticulture (Table 3). The surrounding areas of WA’s Shire of Augusta-Margaret River and City of Busselton in temperate south-western Australia were also identified as highly suitable areas, with some irrigated and perennial horticultural areas (Table 2 and Table 3). SA has the second-largest highly suitable area with 69,756 ha (Figure 3, Table 2), involving some cropping, perennial and intensive horticulture, irrigated cropping, perennial horticulture and seasonal horticulture in temperate southern Australia (Fleurieu Peninsula and the Limestone Coast), as well as a small number of cropping areas in the temperate areas of Eyre and Yorke peninsulas (Figure 3, Table 3). NSW predictions include relatively smaller areas, mostly around temperate eastern Australia (Shoalhaven and South Coast) with the cultivated area encompassing cropping and perennial, seasonal and intensive horticulture (Figure 2, Table 3). Other areas, for example, areas in temperate southern Australia (Glenelg-Southern Grampians in Vic. and West Coast in Tas.), tropical north eastern Australia (Far North and Gladstone–Biloela in Qld) had minor potential establishment areas, which were cropping or irrigated cropping (Figure 2, Table 3).

The majority of moderate suitability and low suitability potential establishment areas in temperate south-western Australia (WA) and tropical north-eastern Australia (Qld) extended along the coast and passed through the whole coastal area of temperate eastern Australia (NSW) to temperate southern Australia (Vic.). Moderate and low areas also cover the eastern coastal area of temperate southern Australia (SA), the northern part of tropical north Australia (NT) and some northern regions of temperate southern Australia (Tas.) (Figure 3, Table 3). Apart from the different habitat suitability of potential establishment areas to cultivation areas, most of the cultivation areas were predicted as not suitable, which was attributable to unfavorable climatic conditions for *T. leucotreta* (Figure 3, Table 2).

### 3.2. Key Climatic Variables Influencing the Predicted Distribution of Thaumatotibia leucotreta

This study identified the important environmental factors determining the potential distribution of *T. leucotreta* in Australia. Based on the optimum model, Bio7 (temperature annual range) had the highest contribution (31.2%) to *T. leucotreta*’s habitat distribution, of the environmental factors assessed, as well as the highest permutation importance (67.3%). Bio4 (Temperature seasonality) (20.9%) and DDA10 (Degree days above 10 °C) (17.9%) also had an important contribution, but with more limited permutation importance (Bio4, 4.8%; DDA108.6%). The lowest contributions were from Bio17 (Precipitation of driest quarter, 4% contribution; 8% importance), and Bio19 (Precipitation of coldest quarter, 0.8% contribution; 1% importance) (Table 4).

The results of the Jackknife test showed that Bio4 (Temperature seasonality) had the highest training gain (Figure 5), as well as the second highest contribution rate. Bio7 (Temperature annual range) had the second highest training gain (Table 4), and Bio17 (Precipitation of driest quarter) and Bio18 (Precipitation of warmest quarter) the least training gain (Figure 5), the least contribution, and relatively low importance (Table 4).

The variable response curve for Bio7 (Temperature annual range) showed that the mean temperature range had an important impact on habitat suitability (Figure 6). The higher temperature ranges significantly decreased the habitat suitability. Habitat suitability increased with Bio4 (Temperature seasonality) but decreased when temperatures became too high. Habitat suitability was relatively proportional to Bio6 (Minimum temperature of coldest month). The variable response curves for Bio13 (Precipitation of wettest month) and Bio17 (Precipitation of driest quarter) identified the extremes for the rainfall in the wettest month and the rainfall in the driest three months for the optimal habitat suitability (Figure 7). Habitat suitability was basically relative to Bio19 (Precipitation of coldest quarter). Suitability increased with Degree days but decreases when the Degree days are too high (Figure 8).

## 4. Discussion

*Thaumatotibia leucotreta* (False codling moth) is polyphagous and able to remain active throughout the year, assisted by a “green bridge” created by alternating susceptible crops. As *T. leucotreta* does not diapause, there is a probability of the pest arriving alive with goods at importing countries as a result of chill coma [28]. There was not a significant increase in interception records for *T. leucotreta* in the EU for a number of different commodities for the period of 2020 to 2022 [25]. This is on account of how the international flower trade has been affected by coronavirus pandemic, COVID-19, which included both a sharp decline in demand for flowers and restrictions in transportation networks [72,73,74]. Hence, Australian biosecurity should focus on inspection of those frequently traded commodities, as well as the less-known route of cut flowers. This research has presented the potential susceptible areas of *T. leucotreta* in Australia with Maxent modeling.

In this study, suitable habitats were identified in not only tropical and sub-tropical regions, but also in temperate areas across Australia. This result changes the risk area profile for *T. leucotreta* in Australia as published by previous studies where the areas were largely focused on tropical and sub-tropical regions [23]. The results do verify those of Loomans [9], who stated that *T. leucotreta*’s habitat preference included not only tropical but also temperate climatic zones. Given that much of Australia’s horticulture is in temperate areas, with warm summers and cold winters (mean temperature of the coldest month between −3 °C–18 °C) [75,76] along with winter-dominant rainfall [72], this study suggests the likely threat from *T. leucotreta* is greater than previously stated.

The predicted establishment of *T. leucotreta* within the Mediterranean climatic regions of Australia is not surprising as they are also consistent with the more recent findings by Loomans [9]. The suitability of Mediterranean climates for *T. leucotreta* was highlighted by its introduction and establishment in Israel [11]. Added to this, established populations of *T. leucotreta* have been found in Mediterranean climatic regions in South Africa [6,32,77]. Another study by Everatt [12] also summarized a range of suitable areas for *T. leucotreta* that included establishments close to the Mediterranean coast in North Africa, the Near East and Europe [6].

This research demonstrated that temperature-related variables generally contributed more to the model than precipitation-related variables. This finding is supported by EPPO [6]. For example, Bio4 (Temperature seasonality) and Bio7 (Temperature annual range) contributed most to the potential establishment of *T. leucotreta*. While Bio6 (Minimum temperature of coldest month) had a lower contribution, it is still relatively important because of its higher training gain for modeling. The higher training gain demonstrated by Bio6 reflects the importance of this variable in determining the model and predicted distribution of *T. leucotreta*. This outcome concurs with EPPO [6] that indicated climatic suitability of *T. leucotreta* strongly relies on the pest’s ability to overwinter and survive under the lowest temperature period of the year. It also demonstrated the close relationship between ‘low-temperature mortality’ of *T. leucotreta* and ‘winter mean monthly lowest average temperature’ [32]. The majority of high suitability areas of Australia predicted in this study had mean annual minimum temperatures from 0–5 °C based on Dawson’s [78] ‘Hardiness Zones of Australia’. In this study, DDA10 (Degree days above 10 °C) is the additional variable that had a substantial contribution to the predicted establishment of *T. leucotreta.* This verified the development threshold of *T. leucotreta* based on its biological characteristics from previous studies [16,32,51].

Precipitation-related factors can also impact on *T. leucotreta* establishment. The response curve of Bio13 (Precipitation of wettest month) and Bio17 (Precipitation of driest quarter) indicated the highest suitability of an area for *T. leucotreta* establishment occurred when the total mean rainfall was relatively low. This implies that high rainfall is not as favorable to *T. leucotreta* establishment. This is similar to Daiber [79,80] who reported that heavy or frequent precipitation will have a negative impact on *T. leucotreta* populations as heavy rainfall has been shown to kill *T. leucotreta* eggs. However, low rainfall does not mean an absence of rainfall, as extremely dry conditions have been reported to have a negative impact on *T. leucotreta* development, with drought likely to limit population growth [6]. These results further support the finding of EPPO [6] regarding humidity as an influential factor in the life cycle of *T. leucotreta*, as the pest cannot avoid drought by passing into diapause. Similarly, Daiber [79] identified that low humidity does not favor egg or pupae survival.

According to the previous findings, *T. leucotreta* is recognized as a tropical pest thriving in warm and moist conditions, with a requirement for humidity from precipitation. In addition, host plants are an essential requirement for *T. leucotreta* establishment and spread. The findings were confirmed in this study, where the entire central region of Australia’s desert zone [81] has no cropping. This closely matches the current distribution in Africa where the pest is only established in southern countries but not in northern Africa, which is dominated by desert. Some northern African countries, with a Mediterranean climate and thriving agricultural industries, are producing some crops susceptible to *T. leucotreta.*

The most important finding of this study is the confirmation that temperate areas of Australia are at risk should *T. leucotreta* be introduced. This research identified a number of agricultural habitats that are highly suitable to *T. leucotreta* establishment in Australia. These included the south-west of Western Australia (from latitude 32° S to 35° S with longitude 115° E to 125° E) and the south-eastern part of South Australia (from latitude 36° S to 40° S with longitude 136° E to 140° E), both temperate and Mediterranean climates. Additional high suitability areas were also predicted in the Western Australian Wheatbelt (center from latitude 32° S with longitude 118° E), which are consistent with previous predictions [23].

In addition to climate, Australian land use is also an important factor for consideration, given that the presence of susceptible host material is essential. Extensive areas containing susceptible hosts are present in eastern NSW and south-west WA. In contrast, south-east Qld, east Vic. and south-east SA all have a relatively small proportion of land with suitable hosts. The NT and east Tas. have limited areas with susceptible hosts.

*Thaumatotibia leucotreta* is not able to move long distances independently, so areas growing susceptible perennial hosts will be more suitable than seasonal hosts. However, if there are surrounding alternative susceptible seasonal hosts growing at other times of the year, the risk of establishment will increase, as hosts are available for longer periods. To provide a sensible determination of where this pest might spread, it is necessary to consider both climate suitability and presence of susceptible crops. The overlay map precisely analyses the possibility of establishment of *T. leucotreta* in Australia and indicates that the areas exposed to the highest likelihood of establishment are mostly confined to the states of Western Australia and South Australia.

In Western Australia, there are large areas implicated in the agricultural surrounds across the City of Albany in the Great Southern region (over 63,376 ha). The region around Albany is a significant contributor to Australian agricultural production and includes horticulture and viticulture industries, with susceptible cultivated capsicum, sweet corn, beans, peas and fruits such as avocados, wine grapes as well as some cut flowers [82]. These are all suitable hosts for *T. leucotreta.* The Bunbury region of WA has the second largest susceptible area (approximately 26,294 ha). This region has a diverse agricultural production sector [83], with high-volume production of susceptible hosts including avocados, wine grapes and nectarines [82]. Industry reports show that most of the vegetables in WA are produced year-round, with some fruit production across the seasons [84]. These regions include Plantagenet and the towns of Albany, Manjimup and Pemberton [85]. Given suitable host availability year-round, and that *T. leucotreta* can be active the whole year, where the climate is suitable, it can be assumed that agricultural production around Albany, Margaret River, Manjimup and Bunbury are under threat from *T. leucotreta* establishment.

In South Australia (SA), there are large areas predicted to be highly susceptible to *T. leucotreta*. in the Fleurieu Peninsula (47,386 ha). These areas include Cape Jervis, Victor Harbor and Kangaroo Island, all with high value annual production of grapes and citrus [82]. Limestone Coast was also identified as another large area with high potential for establishment (22,327 ha). This region is one of the primary production areas for viticulture (including high-value wine production) with its fertile soil, rainfall and groundwater resources [82].

Wang et al. [86] suggested considering not only the distribution of host plants but also the presence of natural enemies in the development of SDM predictions. For future studies, the natural enemies of *T. leucotreta* could be considered as one of the variables in predicting the potential establishment and distribution of the pest. Pest biology is an important factor impacting the distribution of pests through the interaction between the pest and its enemies [87]. Natural enemies of *T. leucotreta* include parasitoids and predators. *Trichogrammatoidea cryptophlebiae* Nagaraja, the African egg parasitoid. is one of the natural enemies of *T. leucotreta* and has been used as an effective biological control agent against the pest [14,88,89,90]. Another effective parasitoid is larval, *Agathis bishop* (Nixon) [91]. Major predators include two ants, *Anoplolepis custodiens* (Smith) and *Pheidole megacephala* (Fabricius), both have been proven to attack the larvae and pupae of *T. leucotreta* in citrus orchards [14,92]. Additionally, there are some reported predators of *T. leucotreta* such as the egg predators, *Orius* sp., the larval predator, *Rhynocoris albopunctatus* (Stål) and the egg/larval mite predator *Pediculoides* sp. [14,92,93,94,95].

Lehmann [87] identified that natural enemies impact on the proliferation of insect species. However, predators and parasites used in biological control rarely totally eradicate pests such as *T. leucotreta*. The effectiveness of biological control is sensitive to some abiotic factors [96], parasitoids such as *T*. *cryptophlebiae* are sensitive to broad spectrum chemical pesticides and their improper application [97]. Even so, natural enemies mentioned, such as *P. megacephala*, were introduced into Australia [98] and competition between them and the pest could reduce pest population growth [99] and slow the establishment of *T. leucotreta.*

## 5. Conclusions

This research using Maxent modeling has predicted a different habitat suitability for *T. leucotreta* in Australia compared to previous reports. The outcome not only verified the potential areas from previous research but also discovered and further enlarged the potential areas. These results contribute towards a more accurate assessment of risks posed by *T. leucotreta* to agriculture and horticulture. Potential invasion areas identified in this study may be underpredicted, attributable to insufficient or the absence of data for accurately determining the cold tolerance of *T. leucotreta* [100].

Areas likely to favor the establishment of *T. leucotreta* exist across Australia, in which there are both a suitable climate and cropping. The poor natural dispersal ability of *T. leucotreta* could confine the spread of the pest to the vicinity of cropping areas permitting eradication. The pest’s polyphagous nature could allow it to infest alternating crops within restricted areas. Early detection is therefore key to preventing potential spread.

Large economic losses have been reported from many areas in sub-Saharan Africa and Israel, due to the damage caused by *T. leucotreta.* The pest’s increasingly frequent interception in the EU, before the global outbreak of the coronavirus pandemic, COVID-19, hit the international fresh cut-flower market, highlights the growing threat regarding *T. leucotreta*’s introduction into Australia via imported fresh cut-flowers from Kenya, and the potential for it to establish itself in high-value agricultural production areas in Australia. The findings of this research show the importance of improving offshore and onshore quarantine inspections. To increase inspection vigilance at borders, the next step should consider not only cut flowers from Kenya, but also other imported commodities that could be infested with *T. leucotreta* attributable to its polyphagia. It is also imperative to establish a range of effective import treatments, such as cold treatment or fumigation, early warning monitoring systems, along with published eradication plans with trap monitoring around susceptible facilities identified by this study and awareness training for inspectors and the community.

## Figures and Tables

**Figure 1 insects-13-00883-f001:**
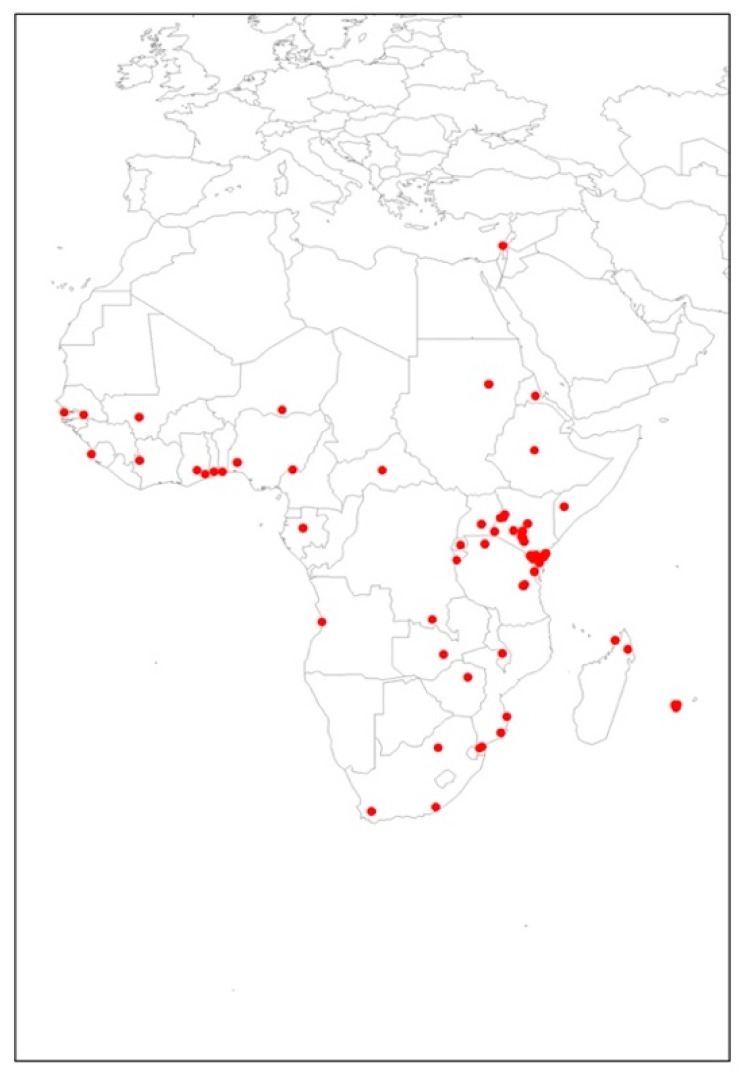
Occurrence points of *Thaumatotibia leucotreta* utilized in the study.

**Figure 2 insects-13-00883-f002:**
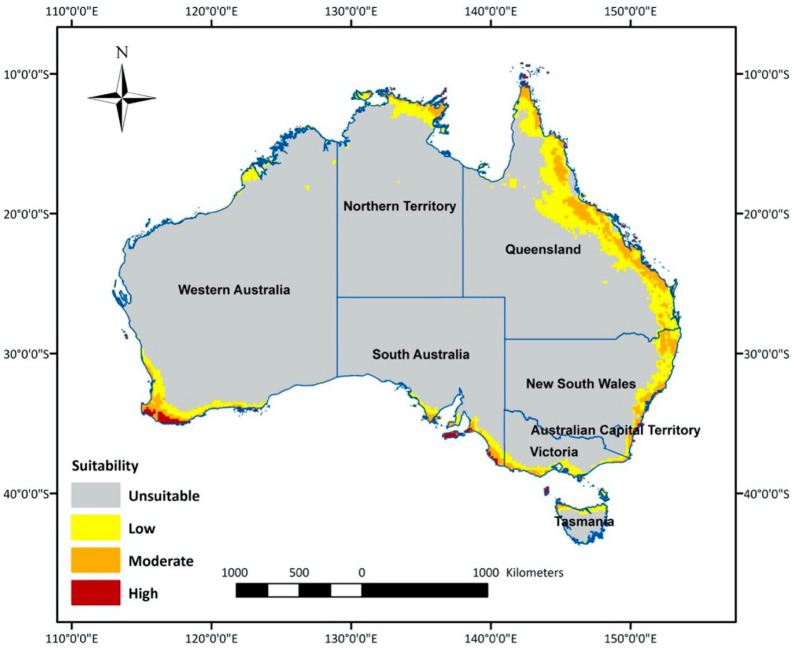
Areas in Australia with a climate suitable (high, medium, low or unsuitable) for *Thaumatotibia leucotreta* establishment.

**Figure 3 insects-13-00883-f003:**
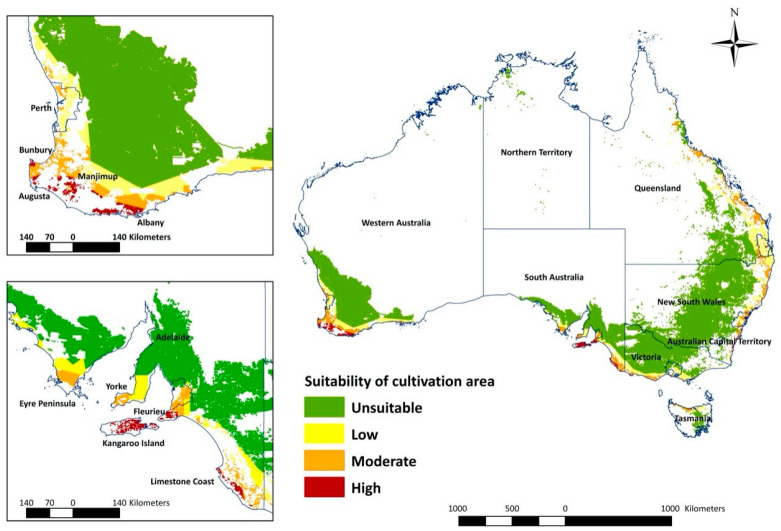
Potential establishment regions for *Thaumatotibia leucotreta* in Australia in relation to suitability of land use (cultivation).

**Figure 4 insects-13-00883-f004:**
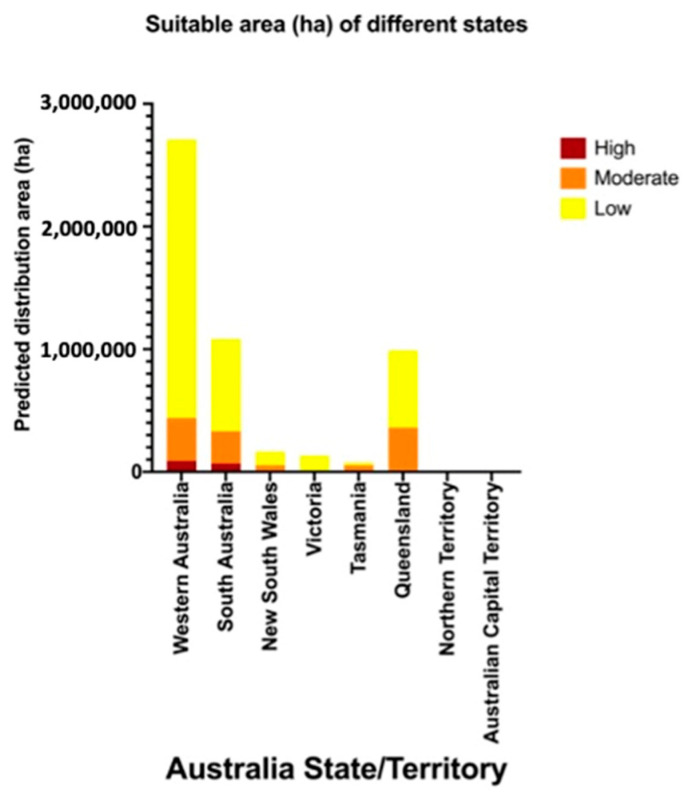
Potential area (ha) for establishment by *Thaumatotibia leucotreta* in each Australian state/territory.

**Figure 5 insects-13-00883-f005:**
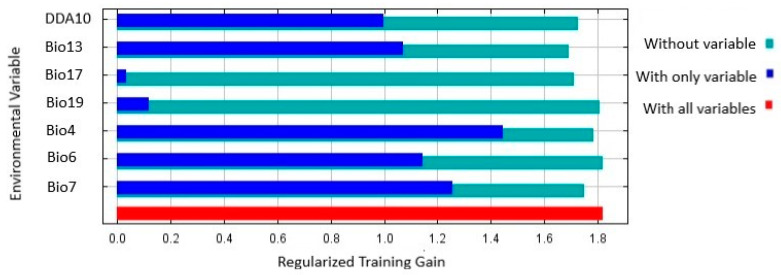
Jackknife test of regularized training gain of the environmental variables and DDA10 used to predict location suitability for *Thaumatotibia leucotreta* in Australia. DDA10 is Degree days above 10 °C.

**Figure 6 insects-13-00883-f006:**
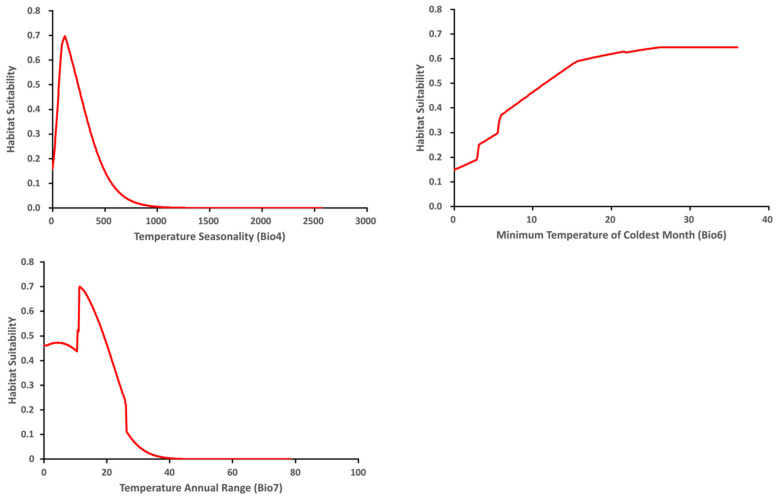
Relationship of habitat suitability response for *Thaumatotibia leucotreta* to three environmental variables (Bio4, Bio6, Bio7) (Units: Temperature °C).

**Figure 7 insects-13-00883-f007:**
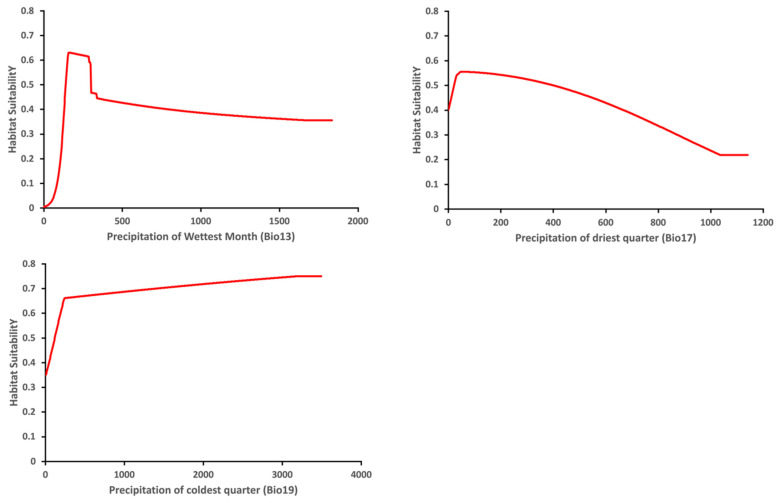
Relationship of habitat suitability response for *Thaumatotibia leucotreta* to three environmental variables (Bio13, Bio17, Bio19) (Units: Precipitation mm).

**Figure 8 insects-13-00883-f008:**
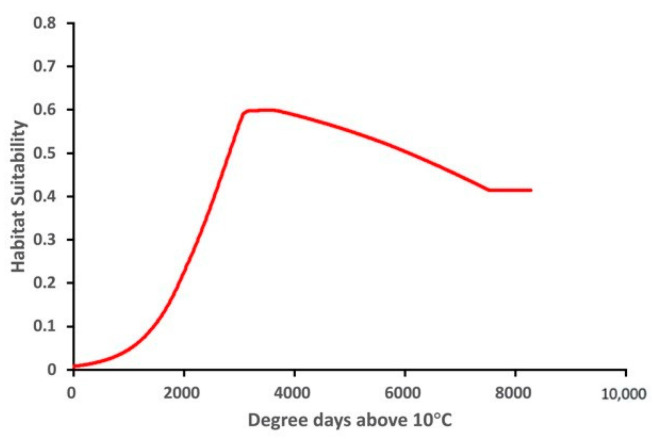
Relationship of habitat suitability for *Thaumatotibia leucotreta* response to Degree days above 10 °C (Units: Degree days dd).

**Table 1 insects-13-00883-t001:** Bioclimatic variables from the WorldClim dataset, plus degree days, assessed to determine suitability for inclusion in *Thaumatotibia leucotreta* predictive modeling. DDA10 is Degree days above 10 °C.

Variable Code	Variable Title	Unit	Modeling Used in This Study
Bio1	Annual mean temperature	°C	
Bio2	Mean diurnal range	°C	
Bio3	Isothermality	%	
Bio4	Temperature seasonality	°C	√
Bio5	Maximum temperature of warmest month	°C	
Bio6	Minimum temperature of coldest month	°C	√
Bio7	Temperature Annual Range	°C	√
Bio8	Mean temperature of wettest quarter	°C	
Bio9	Mean temperature of driest quarter	°C	
Bio10	Mean temperature of warmest quarter	°C	
Bio11	Mean temperature of coldest quarter	°C	
Bio12	Annual Precipitation	mm	
Bio13	Precipitation of Wettest Month	mm	√
Bio14	Precipitation of Driest month	mm	
Bio15	Precipitation seasonality (coefficient of variation)	%	
Bio16	Precipitation of Wettest quarter	mm	
Bio17	Precipitation of Driest quarter	mm	√
Bio18	Precipitation of Warmest Quarter	mm	
Bio19	Precipitation of Coldest quarter	mm	√
DDA10	Degree days above 10 °C	dd	√

**Table 2 insects-13-00883-t002:** Summary of potential establishment areas (ha) for *Thaumatotibia leucotreta* in each Australian state/territory.

	Suitability	High	Moderate	Low	Unsuitable
State/Territory					
**Western Australia**		89,670	348,652	2,271,156	41,408,183
**South Australia**		69,766	262,205	750,311	13,348,525
**New South Wales**		530	53,004	112,464	25,191,545
**Victoria**		81	9564	123,151	10,449,260
**Tasmania**		52	52,179	23,382	152,890
**Queensland**		45	360,547	628,732	4,414,677
**Northern Territory**		3	3	149	37,673
**Australian Capital Territory**		0	0	0	553
**Grand Total**		160,147	1,086,154	3,909,346	95,003,307

**Table 3 insects-13-00883-t003:** Types of cultivation located in different regions of Australia that show potentially high suitability for *Thaumatotibia leucotreta* establishment.

Regions of State/Territory	Area (ha)	Cropping	Perennial Horticulture	Seasonal Horticulture	Production from Irrigated Agriculture and Plantations	Irrigated Cropping	Irrigated Perennial Horticulture	Irrigated Seasonal Horticulture	Intensive Horticulture
Albany (WA)	63,376	√	√	√		√	√	√	√
Manjimup (WA)	22,162	√	√	√	√	√	√	√	√
Augusta-Margaret River-Busselton (WA)	4132						√		
Fleurieu-Kangaroo Island (SA)	47,386	√	√			√	√	√	√
Limestone Coast (SA)	22,328	√	√			√	√	√	
The Eyre Peninsula and South West (SA)	52	√							
Shoalhaven (NSW)	209	√	√	√					√
South Coast (NSW)	321	√	√	√		√			√
Glenelg-Southern Grampians (Vic.)	81	√							
West Coast (Tas.)	52					√			
Far North (Qld)	36					√			
Gladstone-Biloela (Qld)	9	√							
Daly-Tiwi-West Arnhem (NT)	3						√		

**Table 4 insects-13-00883-t004:** Percent contribution and permutation importance of environmental variables used to predict location suitability for *Thaumatotibia leucotreta* in Australia.

Variable Code	Variable Title	Unit	Percent Contribution (%)	Permutation Importance (%)
Bio7	Temperature annual range	°C	31.2	67.3
Bio4	Temperature seasonality	°C	20.9	4.8
DDA10	Degree days above 10 °C	dd	17.9	8.6
Bio13	Precipitation of wettest month	mm	13.3	9.7
Bio6	Minimum temperature of coldest month	°C	11.8	0.6
Bio17	Precipitation of driest quarter	mm	4.0	8.0
Bio19	Precipitation of coldest quarter	mm	0.8	1.0

## Data Availability

The datasets generated during and/or analyzed during the current study are available from the corresponding author on reasonable request.

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
