# Peer review of "Evaluation of the Likelihood of Establishing False Codling Moth (Thaumatotibia leucotreta) in Australia via the International Cut Flower Market"

_insects, 2022, doi:10.3390/insects13100883_

Round 1

Reviewer 1 Report

Abstract:

Is the Kenya only country producing rose and having T. leucotreta, I recommend to change first sentence since in this paper, focus is given to establishment whenever pest originate from (like from South Africa etc..)

In the abstract, do not use formulation “ It is hoped…”, instead give statement following conclusions.

Introduction:

Generally is well explained, still more information about economic impact of T. leucotreta in established area would improve text and give more information to the reader.

Lines:

34 – tipfeller

61-62 the sentence is not clear for the reader

78-79 – state the quarantine list from EPPO where the T. leucotreta has been added

88 and 92, use one name for the pest, either FCM or T. leucotreta during whole paper

Materials and methods:

Methods are well explained for the reader

Results are well explained

Reviewer 2 Report

Straight forward and well presented manuscript on using MaxEnt to discuss spread of T. leucotreta in Australia. Interesting route of entry with cut flowers and expansion to other cropping systems. Good use of discussion of the areas of suitability along with their cropping systems. Please do a thorough read-through of the manuscript for editorial issues throughout. For example, the first sentence of section 4.4 is worded oddly. Also, consider combining figures 6-8, as comparing these visually in one figure may be beneficial to the reader.

Reviewer 3 Report

The authors evaluated the habitat suitability and risk assessment of T. leucotreta in Australia. However, the authors did not take the potential natural enemies of  T. leucotreta into consideration as they are one of the major factors affecting the distribution and establishment of T. leucotreta. I suggest that the authors should improve the discussion section by adding currently known potential natural enemies.

Reviewer 4 Report

This manuscript (MS) examines the assessment of the likelihood of establishment of T. Leucotreta in Australia via the international cut flower market.  This pest is a serious threat to Australian agriculture. I think this MS is very valuable and important for a given area. This study followed the structure of the basic thesis well, and studied an interesting topic. The introduction part needs some improvements. An English spell check is suggested. The authors did not used the correct template that is provided for MS, the text is not aligned. Also, the references aren’t arranged according to the instructions for authors. I suggest to check the instructions for authors very carefully. 
